# Autotaxin Has a Negative Role in Systemic Inflammation

**DOI:** 10.3390/ijms23147920

**Published:** 2022-07-18

**Authors:** Ioanna Nikitopoulou, Aggeliki Katsifa, Paraskevi Kanellopoulou, Edison Jahaj, Alice G. Vassiliou, Zafeiria Mastora, Ioanna Dimopoulou, Stylianos E. Orfanos, Vassilis Aidinis, Anastasia Kotanidou

**Affiliations:** 11st Department of Critical Care Medicine & Pulmonary Services, GP Livanos and M. Simou Laboratories, National and Kapodistrian University of Athens Medical School, Evangelismos Hospital, 10676 Athens, Greece; joannaniki@gmail.com (I.N.); alvass75@gmail.com (A.G.V.); sorfanos@med.uoa.gr (S.E.O.); 2Institute of Fundamental Biomedical Research, Biomedical Sciences Research Center Alexander Fleming, 16672 Vari, Greece; agkatsifa@yahoo.com (A.K.); kanellopoulou@fleming.gr (P.K.); aidinis@fleming.gr (V.A.); 31st Department of Critical Care Medicine & Pulmonary Services, National and Kapodistrian University of Athens Medical School, Evangelismos Hospital, 11527 Athens, Greece; edison.jahaj@gmail.com (E.J.); zafimast@yahoo.gr (Z.M.); idimo@otenet.gr (I.D.)

**Keywords:** sepsis, inflammation, autotaxin, lysophosphatidic acid, endotoxemia, LPS

## Abstract

The pathogenesis of sepsis involves complex interactions and a systemic inflammatory response leading eventually to multiorgan failure. Autotaxin (ATX, ENPP2) is a secreted glycoprotein largely responsible for the extracellular production of lysophosphatidic acid (LPA), which exerts multiple effects in almost all cell types through its at least six G-protein-coupled LPA receptors (LPARs). Here, we investigated a possible role of the ATX/LPA axis in sepsis in an animal model of endotoxemia as well as in septic patients. Mice with 50% reduced serum ATX levels showed improved survival upon lipopolysaccharide (LPS) stimulation compared to their littermate controls. Similarly, mice bearing the inducible inactivation of ATX and presenting with >70% decreased ATX levels were even more protected against LPS-induced endotoxemia; however, no significant effects were observed upon the chronic and systemic transgenic overexpression of ATX. Moreover, the genetic deletion of LPA receptors 1 and 2 did not significantly affect the severity of the modelled disease, suggesting that alternative receptors may mediate LPA effects upon sepsis. In translation, ATX levels were found to be elevated in the sera of critically ill patients with sepsis in comparison with their baseline levels upon ICU admission. Therefore, the results indicate a role for ATX in LPS-induced sepsis and suggest possible therapeutic benefits of pharmacologically targeting ATX in severe, systemic inflammatory disorders.

## 1. Introduction

Sepsis, the leading mortality cause in the intensive care unit (ICU) worldwide [1], is a systemic reaction to infection which can eventually lead to multiorgan failure. During the activation of inflammatory responses, multiple pathways are involved, and complicated cellular functions are affected. The management of sepsis is imperative due to the long-term consequences along with the significant burden for the healthcare system [2].

Autotaxin (ATX; ENPP2) is a secreted lysophospholipase D that is present in most biological fluids and is largely responsible for the extracellular production of lysophosphatidic acid (LPA) from the hydrolysis of lysophosphatidylcholine (LPC) [3]. LPA is a bioactive phospholipid that exerts multiple biological activities in most cell types through its G-protein-coupled receptors (GPCRs; LPAR1-6), which are involved in several signal transduction pathways [4]. A pathologic role for the ATX/LPA axis has been attributed in several chronic inflammatory diseases and cancers [5,6]. Increased ATX expression has been demonstrated in asthma [7], idiopathic pulmonary fibrosis (IPF) [8,9] and COVID-19 [10], where it correlated with the components of the cytokine storm.

However, much less is known about the involvement of ATX/LPA in severe systemic inflammation. In sepsis patients, serum LPC, the substrate of ATX concentration, was found to be lower compared to that in healthy controls [11]. Additionally, serial LPC measurements were shown to be useful in predicting 28-day mortality in septic patients [12]. As proven recently, ATX is also upregulated in liver failure syndromes, and monocyte proinflammatory cytokine production is induced by LPA [13]. Therefore, in this study, we investigated a possible role for ATX in systemic inflammation. For that reason, the effect of ATX genetic deletion was studied upon lipopolysaccharide (LPS) administration in mice. Moreover, ATX levels were determined in samples from ICU patients who developed sepsis.

## 2. Results

### 2.1. ATX Haploinsufficiency Protects Mice from LPS-Induced Sepsis

In this study, we aimed to assess the role of ATX in systemic inflammation. Heterozygous knockout *Enpp2^+/−^* mice were studied, since *Enpp2^−/−^* mutant embryos show embryonic lethality due to abnormal vascular development and neural defects [14]. Heterozygous ATX knockout mice (*Enpp2^+/−^* or ATX^df/+^) were obtained upon intercrossing the ATX conditional knockout (*Enpp2^n/n^*) mice with a transgenic mouse strain expressing the Cre recombinase driven by the CMV promoter [15]; no phenotypic malformations were observed. However, these mice produce 50% reduced serum ATX levels and ATX activity levels (Appendix A, [14,16,17]), accompanied by a 50% decrease in plasma LPA levels.

In order to study whether ATX is involved in the pathogenesis of mouse endotoxemia, the mice were injected intraperitoneally with LPS (from Escherichia coli dissolved in normal saline), an endotoxin present in the outer membrane of Gram-negative bacteria that induces an acute phase inflammatory response mimicking several of the initial clinical features of sepsis [18]. As recently described, LPS administration in these settings results in the upregulation of the serum levels of TNFα and IL-6 [19]. The corresponding mRNA levels of TNF, IL-1b and IL-6 in the liver, spleen and lung tissues were also found to be increased (Appendix A), as previously shown [20]. Normal saline alone was administered to littermate mice; the survival and health status of mice were monitored every 4 h during the light period. Haploinsufficient ATX^df/+^ mice were found to have an increased survival rate compared to their littermate controls (Figure 1), indicating that ATX is involved in the pathogenesis of LPS-induced sepsis.

To avoid the potential developmental effects of the ATX genetic deletion and to obtain even lower ATX levels, *Enpp2^n/n^* mice [14] were crossed with the *R26-Cre^ERT2^* mice that carry a Cre recombinase-estrogen receptorT2 (Cre^ERT2^) allele. Tamoxifen (Tmx), a synthetic estrogen antagonist, serves as a useful tool to study gene functions by enabling the nuclear translocation of Cre and the recombination of floxed DNA alleles. Tmx was administered to *R26-Cre^ERT2^/Enpp2^n/n^* and control littermate mice, and the induced R26Cre-ERT2-mediated *Enpp2* recombination resulted in >70% decreased ATX activity levels (Appendix A, [21]). The mice bearing the inducible inactivation of ATX (*R26-Cre^ERT2^/Enpp2^n/n^* mice) exhibited greater protection against LPS-induced endotoxemia compared to the control mice (Figure 2), further supporting the ATX involvement in sepsis.

### 2.2. Excess Circulating ATX Levels Have No Effect on LPS-Induced Sepsis

To investigate whether increases in serum ATX levels could exacerbate LPS-induced sepsis, transgenic mice overexpressing ATX in the liver under the control of the human α1-antitrypsin promoter (a1t1), resulting in ~200% increased serum ATX/LPA levels [8,22,23,24], were used. LPS was administered to homozygous transgenic mice (Tga1t1*Enpp2*) and to their littermate controls, and the overall survival was studied. Chronically elevated serum ATX levels did not seem to alter LPS-induced mortality in comparison to the control mice (Figure 3), suggesting that long-term systemic increases in ATX levels do not significantly affect sepsis.

### 2.3. The Ubiquitous Genetic Deletion of Lpar1 and Lpar2 Has No Significant Effect on LPS-Induced Endotoxemia

The possible involvement of LPAR1 and LPAR2 in systemic inflammation was then examined through the administration of LPS to *Lpar1^−/−^*, *Lpar2^−/−^* mice and their WT littermates. Since C57BL/6J *Lpar1*^−*/*−^ mice are embryonically lethal, Lpar1^−/−^ mice were bred and maintained on a mixed C57BL/6J/129 genetic background [25]. The deletion of *Lpar1* tended to improve survival, albeit with no statistical significance (Figure 4A). No effect in survival was observed from the genetic deletion of *Lpar2,* ruling out its involvement in LPS-induced sepsis (Figure 4B).

### 2.4. Elevated ATX Levels in Critically Ill Patients with Sepsis

Having identified the importance of the ATX-LPA axis, we then focused on possible fluctuations in patients’ systemic levels. The demographics and baseline characteristics of the patients included in the study are summarized in Table 1. Overall, 26 ICU patients aged 45 ± 20 years were recruited at Evangelismos Hospital.

ATX levels were measured repetitively in the sera of ICU patients ≤ 24 h after admission as well as at sepsis development. As shown in Figure 5, the ATX peripheral concentrations in sepsis were found to be elevated compared to the baseline (i.e., non-sepsis) values. The increased levels of ATX during sepsis strongly support its role in disease pathophysiology.

## 3. Discussion

The pathophysiology of sepsis involves the deregulation of the immune system and the production of pro-inflammatory mediators. Lipid mediators have been recently proposed to mediate the immune response, thus playing a role in critical illness mechanisms [26]. In the present study, we genetically identified a role of the ATX-LPA axis in LPS-induced endotoxemia in mice. More specifically, we demonstrated that haploinsufficient ATX^df/+^ mice, producing 50% reduced serum ATX levels (Appendix A, [14,16,17]), show increased survival compared to their littermate controls (Figure 1). Similarly, mice bearing the Tmx-inducible >70% inactivation of ATX (Appendix A) [21] were even more protected against LPS-induced endotoxemia compared to their control littermate mice (Figure 2).

LPS administration in mice is a widely used animal model to study sepsis, which partially mimics some of the initial clinical features in humans, including the serum increases of pro-inflammatory cytokines, such as TNFα, IL-6 and IL-1b [27,28,29]. The severity and timing of LPS-induced sepsis in mice vary depending on the source, amount and route of administered LPS, as well as the mouse strain, the local genetic drift and the health status and microbiome load of the local animal house [27,28,29]. In this context, the conditions used in this study were selected upon extensive local testing over the years, focusing on 48 h survival.

Therefore, the genetic deletion of ATX, both in an obligatory as well as in an inducible way, resulted in a reduction of LPS-induced lethality, indicating that ATX, and therefore LPA, play a pathogenic role in the development of LPS-induced experimental sepsis. In agreement, the pharmacologic inhibition of ATX in mice reduced systemic (IP) LPS-induced serum TNFα and IL-6 levels, as well as the mRNA levels of different proinflammatory mediators in the mouse brain, such as TNFα, IL-1β, IL-6, iNOS and CXCL2 [30].

However, transgenic ATX overexpression from the liver and the ensuing serum 200% ATX increases [22] did not exacerbate LPS-induced sepsis and lethality (Figure 3). Of note, the increased systemic ATX levels in the same ATX transgenic mouse were previously shown not to significantly affect the development of modelled chronic inflammatory diseases, such as bleomycin-induced pulmonary fibrosis [8], transgenic TNF-driven inflammatory arthritis [23] or experimental autoimmune encephalomyelitis [24]. On the contrary, the transgenic ATX overexpression and the systemic serum ATX increases did exacerbate the development of CCl4-induced hepatitis [31], suggesting that systemic ATX can affect metabolic active tissues, such as the liver [31,32] or the muscle [33], and not just inflamed tissues.

The genetic deletion of Lpar1 tended to improve survival after LPS administration, albeit in a non-statistically significant way (Figure 4A). This trend is consistent with previous findings revealing that LPAR1-deficient mice show attenuated lung vascular permeability induced by bleomycin [34]. Moreover, the pharmacologic inhibition of LPAR1 with ki16425 has been reported to reduce the severity of LPS-induced abdominal inflammation and organ damage [35]. LPAR1 antagonism reduced inflammatory markers, as well as mortality [35]. Lpar2 genetic deletion, on the other hand, did not seem to have an impact on survival upon LPS-induced endotoxemia (Figure 4B. Moreover, the pharmacologic inhibition of LPAR5 with AS2717638 has been recently reported to attenuate IP LPS-induced serum cytokine levels as well as inflammatory markers in the mouse brain [30]. Therefore, both LPAR5 and LPAR1, but not LPAR2, mediate to some extent the pathogenic effects of ATX/LPA in experimental sepsis.

To translate the animal model findings into the human disease, increased ATX levels were observed in septic patients in comparison with their levels upon ICU admission (Figure 5), suggesting a possible involvement of ATX in human sepsis too. However, in the present study, no statistical correlation between ATX levels and any clinical endpoints and/or scores was observed, possibly due to the relatively low number of patients. Additional clinical studies will be needed to delineate the exact role of ATX in the pathogenesis of sepsis in humans. In accordance, increased ATX activity has been reported in septic patients who did not survive up to 30 days following discharge, corelating with platelet count and the ratio of angiopoietin-2/1 (Ang-2/1) [36], indicating an association of ATX, platelets and endothelial dysfunction. Increased serum ATX levels have been recently reported in severe COVID-19 patients, also correlating with the markers of endothelial dysfunction (sP-sel, sICAM) [10], further enforcing the correlation between ATX/LPA and endothelial damage [37]. In addition, it has been recently shown that increased serum ATX levels are associated with severe acute respiratory distress syndrome (ARDS) and an unfavorable outcome [38]. This study showed that increased ATX levels are correlated with inflammatory and fibrotic biomarkers, denoting the role of ATX as a predictor in ARDS. Of note, ATX was found to add value in predicting the outcome of ARDS, both alone and in combination with SOFA, APACHE II and PaO_2_/FiO_2_. Our observations also agree with reports on decreased LPC and increased ATX expression and LPA concentration in acute-on-chronic liver failure [13]. In particular, LPA production was assigned a pivotal role in the regulatory phenotype and function of monocytes [13]. The treatment of CD14+ cells with LPA increased monocyte TNF-α production, thus supporting the notion that LPA has a role in the reprogramming of monocytes and the modulation of immune functions.

Overall, a pathogenic role for ATX in LPS-induced experimental sepsis was indicated via the genetic, obligatory or inducible deletion of ATX. Moreover, the development of sepsis in human ICU patients was accompanied by serum increases of ATX, further suggesting a role for ATX/LPA in human sepsis. The establishment of ATX as a therapeutic target in pulmonary fibrosis has led to the development of numerous ATX inhibitors [39,40], which could also be proven to be useful for sepsis management.

## 4. Materials and Methods

### 4.1. Mice

The mice were bred at the animal facilities of the Biomedical Sciences Research Center ‘Alexander Fleming’, under specific pathogen-free conditions, at 20–22 °C and under 55 ± 5% humidity and a 12 h light–dark cycle; food and water were provided ad libitum. The mice were bred and maintained in a C57BL/6 genetic background for more than 10 generations, except for Lpar1^−/−^ mice, which were bred in a mixed C57BL/6J/129 genetic background. All of the experimentation was approved by the Institutional Animal Ethical Committee (IAEC) of the Biomedical Sciences Research Center ‘Alexander Fleming’, as well as by the Veterinary Service of the governmental prefecture of Attica, Greece (approval protocol number 3765/2011). The study was conducted in compliance with the European Union Directive 2010/63/EU on animal experimentation. All efforts were made to minimize animal distress and suffering. The health status of the mice was monitored daily. The mice were euthanized at predetermined time-points under deep anesthesia by exsanguination. The generation and genotyping protocols for Enpp2^n/n^ [14], R26Cre^-ERT2^ [41], Tga1t1-hEnpp2 [22], Lpar1^−/−^ [25] and Lpar2^−/−^ [42] mice have been previously described.

### 4.2. Tamoxifen Treatment

Tamoxifen (Tmx; Sigma T5648, St. Louis, Missouri, USA) was dissolved in a corn oil/ethanol (9/1) mixture at 45 mg/mL. Tmx was administered by oral gavage (Per Os, PO; 180 mg/kg), as previously described [43]. The control groups received corn oil.

### 4.3. Survival Study

LPS (Escherichia coli O111:B4, Sigma-Aldrich, Schnelldorf, Germany) was injected intraperitoneally (20 mg/kg), as previously described [19], in ATX^df/+^, *R26 Cre^ERT2^/Enpp2^n/n^*, *Enpp2*-Tg, *Lpar1^−/−^* and *Lpar2^−/−^* mice. Kaplan–Meier survival analysis was performed.

### 4.4. ATX Activity Assay

ATX/LysoPLD activity was measured using the TOOS activity assay. Hydrogen peroxide serves as the oxidizing agent, and, in the presence of horseradish peroxidase, it reacts with TOOS (*N*-ethylN-(2-hydroxy-3-sulfopropyl)-3-methylaniline) and 4-AAP (aminoantipyrene) to form a pink quinoneimine dye which absorbs at 555 nm. 1× LysoPLD buffer (100 mMTris-HCl pH 9.0, 500 mM NaCl, 5 mM MgCl_2_, 5 mM CaCl_2_, 60 μM CoCl_2_, 1 mM LPC) was pre-incubated at 37 °C for 30 min. Plasma samples were incubated with 1x LysoPLD buffer at 37 °C for 4 h. At the end of the incubation, a color mix (0.5 mM 4-AAP, 7.95 U/mL HRP, 0.3 mM TOOS, 2U/mL choline oxidase in 5 mM MgCl_2_/50 mMTris-HCl pH8.0) was prepared and added to each well. Absorbance (A) was measured at 555 nm every 5 min for 20 min. For each sample, the absorbance was plotted against the time, and the slope (dA/min) was calculated for the linear portion of each reaction. ATX activity was calculated according to the following equation: Activity (U/mL) = (μmol/min/mL) = [dA/min(sample)—dA/min(blank)] × Vt/(e × Vs × 0.5), where Vt: total volume of reaction (mL), Vs: volume of sample (mL), e: millimolar extinction coefficient of quinoneimine dye under the assay conditions (e = 32.8 μmol/cm^2^) and 0.5: the moles of the quinoneimine dye produced by 1 mol of H_2_O_2_.

### 4.5. Real Time RT-PCR

The total RNA was isolated from the tissues using the Tri Reagent in accordance with the manufacturer’s instructions. Reverse transcription was performed with the M-MLV reverse transcriptase (Invitrogen 2334659, Waltham, MA, USA) at a final volume of 20 μL. PCR reaction mixtures were prepared using the SYBR select master mix (Applied Biosystems 01196818, Waltham, MA, USA), followed by quantitative PCR on a BioRad CFX96 Touch™ Real-Time PCR Detection System (Bio-Rad Laboratories Ltd., Hercules, CA, USA). The following primer pairs were used: Beta2-microglobulin (b2m) Fwd: 5′-TTC TGG TGC TTG TCT CAC TGA-3′; Rev: 5′-CAGTAT GTT CGG CTT CCC ATTC-3′; Interleukin 6 (IL-6) Fwd: 5′- CAA AGC CAG AGT CCT TCA GAGA-3′; Rev: 5′-TGT GAC TCC AGC TTA TCT CTT GG-3′; Interleukin 1b (Il-1b) Fwd: 5′-GCA ACT GTT CCT GAA CTC AACT-3′; Rev: 5′-ATC TTT TGG GGT CCG TCA ACT-3′; TNF Fwd: 5′- CCT GTA GCC CAC GTC GTAG-3′; Rev: 5′-GGG AGT AGA CAA GGT ACA ACCC-3′. B2m served as the internal control to normalize the amount of loaded cDNA.

### 4.6. Human Sample Collection and Atx Measurement

All of the studies were performed in accordance with the Helsinki Declaration principles. Informed consent was obtained from all of the individuals or the patients’ next-of-kin for severe cases. The study was approved by the Evangelismos Hospital Research Ethics Committee (#80 1/2/10). The patients’ epidemiological, clinical and experimental data are summarized in the corresponding table. Venous blood (3 mL) was collected within the first 24 h after ICU admission. Autotaxin serum concentrations were measured using an enzyme-linked immunosorbent assay (ELISA), according to the manufacturer’s instructions (R&D Systems Systems Inc., Minneapolis, MN, USA, # DY5255-05).

### 4.7. Statistical Analysis

All data are expressed as the means ± SEMs. The normality and statistical analysis was conducted with Graph Pad Prism 8 (Graph Pad Software, Inc., San Diego, CA, USA), as indicated in each figure legend, along with numbers (n) and statistical thresholds.

## Figures and Tables

**Figure 1 ijms-23-07920-f001:**
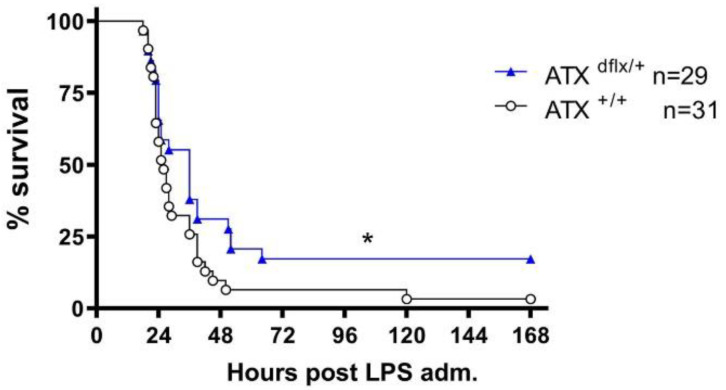
Increased survival rates in LPS-treated heterozygous transgenic AΤΧ knockout mice. Kaplan–Meier survival curves of ATX^df/+^ mice and appropriate littermate controls. Presented results are cumulative from four independent experiments. Differences were tested with the Logrank test. * *p* < 0.05.

**Figure 2 ijms-23-07920-f002:**
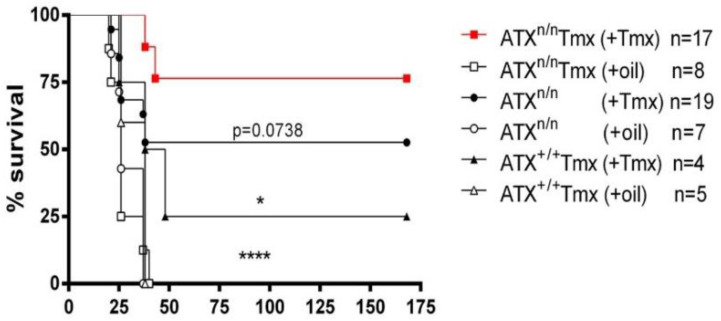
Increased survival rate upon the Tmx-induced genetic inactivation of ATX in mice. Kaplan–Meier survival curves of R26Cre-ERT2/Enpp2^n/n^ and control mice administered with Tmx. * denotes *p* < 0.05 and **** *p* < 0.0001.

**Figure 3 ijms-23-07920-f003:**
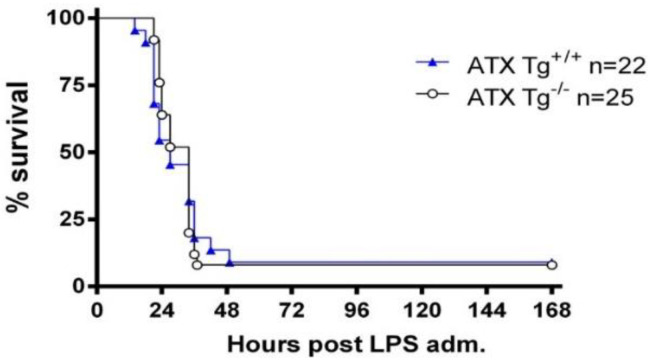
No difference in the survival rates of *Enpp2*-Tg mice overexpressing autotaxin. Kaplan–Meier survival curves of *Enpp2*-Tg mice and appropriate littermate controls. The presented results are cumulative from two independent experiments. No statistically significant differences were observed, as assessed with the Logrank test.

**Figure 4 ijms-23-07920-f004:**
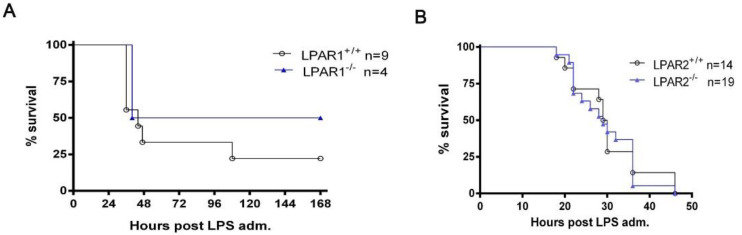
The ubiquitous genetic deletion of LPA receptor 1 or LPA receptor 2 does not significantly affect survival rates after LPS administration. Kaplan–Meier survival curves of LPAR1^−/−^ (**A**), LPAR2^−/−^ (**B**) and their appropriate littermate controls. Differences were tested with the Logrank test.

**Figure 5 ijms-23-07920-f005:**
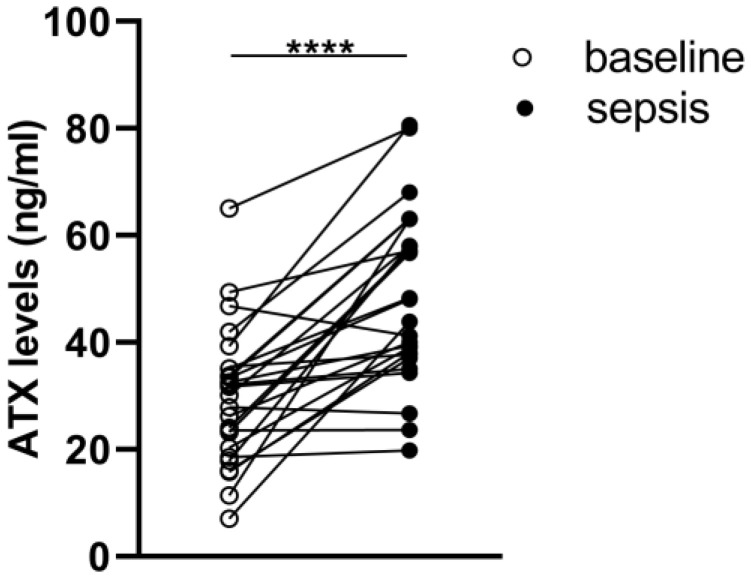
Increased ATX concentrations in patients with sepsis in comparison to their baseline levels ≤24 h after ICU admission. ATX levels were measured using a commercially available ELISA assay. Statistical significance was assessed with the Wilcoxon signed rank test for paired data analysis. **** denotes *p* < 0.0001.

**Table 1 ijms-23-07920-t001:** Clinical characteristics and laboratory data of the patients included in the study.

Patient Clinical Characteristics	
Number of patients (*n*)	26
Sex, *n* (%)	Male 22 (85%) Female 4 (15%)
Age (years, median, IQR)	46 (24–58)
Comorbidities, *n* (%)	Hypertension 13 (50%) Diabetes 4 (15%) COPD 9 (35%)
Hospital stay, days (median, IQR)	13 (7–30)
Hospital mortality (% *n*)	15%
PaO2/FiO2, mmHg (mean ± SD)	259 ± 88
APACHE II score (median, IQR)	11 (8–14)
SOFA score (median, IQR)	6 (4–7)
CRP (mg/dL) (median, IQR)	8 (3–24)
White blood cell count (per μL) (median, IQR)	6000 (3800–7000)

CRP: C-reactive protein; APACHE: acute physiology and chronic health evaluation; SOFA: sequential organ failure assessment.

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
