# Peer review of "Autotaxin Has a Negative Role in Systemic Inflammation"

_ijms, 2022, doi:10.3390/ijms23147920_

Round 1

Reviewer 1 Report

Nikitopoulou and colleagues presents draft entitled Autotaxin has a detrimental role in systemic inflammation.

I went carefully for the whole draft and I found it interesting, well structured and well written. I have no majors nor minors regarding introduction section as well as major part of the discussion.

I would like to express my majors that need to be addressed:

The usage dose of LPS is debatable/questionable. LD50 dose with ip route of administration is 10 mg/kg for black six mice. Here, the Authors used 20 mg/kg that is definitely very high. I also do not agree that ip route is better than iv. All the highest rigors say that i.v. LPS administration is the most right approach when studying septic shock. I must to ask the Authors to repeat the mice part of the paper with lower dose of LPS (e.g. 0.1 mg/kg) with intravenously done injection.

I am also concerned about all the survival rates data. Please see, there is almost no differences within the first 36 hours after LPS dosage. It looks like the autotaxin comes into play much later what is in the opposite to the main paper statement. I am sure that addition new iv LPS data might shed a new light onto this issue.

I also feel that the mice work is not done comprehensively. I would like to see the evidence of the activation of innate immune system as well as the cytokine profile of the mice groups. Even basic PCR/WB data would make this paper much stronger. And findings more reliable.

I do not understand the Figure 5. In Table 1, Authors state that 26 patients were included into trial. Later on, the graph containing 18 patients is shown. It is very confusing. Please explain.

Moreover, the novelty of the paper is low. The importance is high, but it has been shown many times that the Authors made the right assumptions and conclusions. But there is nothing added to science. Please refer to nicely written review by the Authors: DOI: 10.1016/j.jaut.2019.102327  . As well as other papers: https://doi.org/10.1155/2017/9173090 ; https://www.jimmunol.org/content/192/3/851;

To sum up. Without adding new data this paper does not meet the criteria for being published in IJMS. However, I am always supportive when good revision work is done. I encourage Authors to do this.  

Reviewer 2 Report

This is an interesting study that employs genetically engineered mice to evaluate the role for Autotaxin (ATX) in a LPS-stimulated model of sepsis in mice.  Several issues impact on the conclusions being drawn and require addressing as follows:

1. Evaluation of ATX under conditions where genetically altered mice are used needs to be shown not simply identified as data not shown.  Suggest to include these data as a supplement.

2. With regards to Figure 3 - It would be interesting to reduce the LPS dose to a point where you can extend the survival period and then ascertain whether there is an impact of ATX over-expression - there seems to be a trend that the ATXTg+/+ mice seem to shift the survival curve slightly to the left under the conditions employed here - perhaps titrate the LPS dose.  A decrease in survival ascertained in ATX-Tg+/+ mice would seem an ideal outcome worth evaluating further.

3.  With regard to the evaluation of ATX levels in human subjects - Difficult to establish cause or consequence from this data set as paired data (from the same patient) is not available.  At best this is suggestive of a role for ATX
Are any correlations possible with the level of ATX and clinical complications?  Particularly looking at organ failure or the extent of dysfunction - such as kidney or liver?  Or even a systemic vascular biomarker like BP?

4. Page 6 - paragraph preceding the conclusion statement - This section of the discussion completely lacks references - many of the remarks here are simply opinion unless substantiated - since this study did not prove any of the statements made here then this needs to be corroborated with literature based evidence.

AT least one paper that may be supportive is shown below from a search using the PUBMed data base:

Inhibition of Autotaxin and Lysophosphatidic Acid Receptor 5 Attenuates Neuroinflammation in LPS-Activated BV-2 Microglia and a Mouse Endotoxemia Model Int J Mol Sci . 2021 Aug 7;22(16):8519.

5. Methods section - aniaml model - LPS (20 mg/kg) adminstered by i.p. injection - In relation to the suggestion to titrate LPS to slow the extent of inflammation and allow ATX over-expression to impact the extent of inflammation and decrease viability - 20 mg LPS/kg seems a high dose that may bring on a very rapid phenotype.

6. Regarding the pairwise comparison using student t-tests; This test can only be used on data that is demonstrated to be parametric in nature- hence a normalcy test must be conducted and the outcome described in order to justify the use of this approach - where data sets are not normal then a more suitable non-parametric statistical test needs to be used.

Round 2

Reviewer 2 Report

The authors have revised the submission according to the reasonable comments of the reviewers.  I agree with the authors that the manuscript is now much stronger and the statistical analyses are now suitably described and applied to the data sets shown.

Many of the changes added into the text for the revised version are warranted and go toward meeting the expectations of the initial review including the addition of data that was previously not shown in the original manuscript (new supplementary figure  S1).

I have no further comments that require attention by the authors and I form the opinion that the revised manuscript is significantly improved.